# The Effects of Aerated Irrigation on Soil Respiration and the Yield of the Maize Root Zone

Zhenzhen Yu [1,2], Chun Wang [1,2,*], Huafen Zou [2], Hongxuan Wang [2], Hailiang Li [2], Haitian Sun [1,2] and Deshui Yu [3]

1   College of Engineering, Heilongjiang Bayi Agricultural University, Daqing 163319, China; yudq1994@hotmail.com (Z.Y.); sht317770691@163.com (H.S.)
2   National Soil Quality Zhanjiang Observation and Experimental Station, South Subtropical Crops Research Institute, Chinese Academy of Tropical Agricultural Sciences, Zhanjiang 524000, China; zhfjh01@163.com (H.Z.); xuan98926496@126.com (H.W.); lihailiang@126.com (H.L.)
3   School of Management, Huazhong University of Science and Technology, Wuhan 430074, China; yds@hust.edu.cn
*   Correspondence: wangchun1963@126.com

**Abstract:** To investigate the effect of aerated irrigation on the soil environment and yield in the root zone of maize, and to provide a basis for the extension of aerated irrigation, a 2-year experiment (2020–2021) was conducted at the Zhanjiang National Soil Quality Observation Experiment Station, with two experimental observations per year (spring-summer and fall-winter) to investigate the effects of aerated irrigation (AI) and non-aerated irrigation (CK) on soil respiration rate, soil temperature, water content, oxygen content, soil bacterial biomass and root biomass. We used partial least square regression analysis (PLSR) to establish the regression equations of soil respiration rate, soil temperature, water content, oxygen content, soil bacterial biomass and root biomass under the two treatments, and the screening of the main soil environmental factors affecting changes in soil respiration rate under aerated irrigation technology. The results showed that, compared with CK, the AI treatment significantly increased the soil respiration rate and soil oxygen content (15.38~17.87% and 18.94~25.17%, respectively), as well as the root biomass and soil bacterial biomass (14.99~19.09% and 35.10~45.59%, respectively), and reduced the soil water content by 5.33~12.71% ($p < 0.05$). The effects of different treatments on soil temperature were not significant. The mean fruit yield with AI treatment was also 7.16~20.51% higher ($p < 0.05$) than that with CK, and the stem thickness and leaf area of maize plants were significantly increased (9.31~17.06% and 8.68~15.20%, respectively ($p < 0.05$)). The regression fitting results showed that the soil respiration rate is quadratic polynomial negatively correlated with soil temperature, water content and soil oxygen, and the power function is positively correlated with root biomass and bacterial biomass under the two treatments. The variable importance for projection (VIP) values of the PLSR model showed a soil temperature VIP = 1.51, soil oxygen content VIP = 1.42 and root biomass VIP = 1.40, demonstrating that aerated irrigation technology can drive soil respiration rate by changing soil oxygen content and root biomass. Furthermore, the improvements in soil aeration conditions and respiration with AI appeared to facilitate the improvement in fruit yields, which also suggests the economic benefits of AI.

**Keywords:** aerated irrigation; soil respiration; soil oxygen; yield; maize

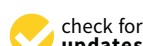



## 1. Introduction

Agriculture, as the primary consumer of water resources around the world, is increasingly being affected by the demands of other sectors of society and threatened by anthropogenic climatic change [1]. Subsurface drip irrigation (SDI) has shown great potential for improving water use efficiency, reducing irrigation water application, and minimizing the potentially negative environmental effects of irrigation. However, oxygen deficiency in the soil caused by sustained wetting fronts under SDI can negatively impact root aeration [2,3].

Related studies have shown that the tomato yield decreased by 32% and 2.5%, respectively, when the drip head burial depth changed from 20 to 40 cm [4,5]. Moreover, crop roots preferentially grow near the drip head [6], which exacerbates the harm of soil hypoxia to crop roots near the drip head; on the other hand, the higher frequency of subsurface drip irrigation [7] will intermittently lead to an increase in soil water content, which increases the degree of tortuosity of soil oxygen transport paths [8] and reduces the availability and diffusivity of soil oxygen [9].

Aerated irrigation (AI) has emerged as a method to mitigate hypoxic conditions; it is defined as the delivery of aerated water directly to the root zone by subsurface drip irrigation (SDI) [10–12]. With the use of AI, substantial quantities of oxygen both in the gaseous phase and dissolved in water can be delivered via subsurface pipes and emitters to the root zone. Many studies have shown the advantages of AI for crop growth and yield potentials [13–16]. Furthermore, on the basis of SDI, AI further improves water use efficiency [17–19]. At the macro-level, sustainable irrigation developed from AI and SDI to balance the supply of soil water, oxygen, nutrients, and agrochemicals may provide a future direction for irrigation [20]. Research on AI should pay more attention to variations in the soil's micro-environment, including soil microorganisms [21], enzymes [22], oxygen [23,24], greenhouse gas emissions [25,26] and root morphology [27]. Therefore, this research further quantifies the effects of AI on a subset of the soil's micro-environment factors, including soil abiotic factors (soil respiration, oxygen, water, and temperature) and soil biotic factors (bacterial biomass and root biomass).

Research on the effects of topsoil aeration on soil respiration and soil oxygen diffusion have shown significant effects of soil crustings and compaction [28–30]. Bhattarai [17] revealed the benefits of AI on soil oxygen content. However, the mechanism of the effect of aerated irrigation technology on the change of soil respiration rate is mostly focused on the analysis of the effect of a single factor or two factors in the soil environment, such as the relationship between soil respiration, oxygen content and soil heat and water. Research on soil respiration and factors that control these fluxes has mostly focused on forest and grassland ecosystems [31–33]. Fewer studies have concentrated on the variation in soil respiration in agricultural field ecosystems or in greenhouses.

In this experiment, the diurnal variations in soil respiration rate, soil temperature and soil oxygen content, as well as the seasonal variations in soil respiration rate, soil temperature, soil water content, soil oxygen content, soil bacterial biomass, root biomass, maize growth and yield were analyzed for both the AI and CK treatments. The objectives of this research were: (1) to determine if there are significant differences in soil environment and maize growth values under AI conditions compared with those under CK conditions; (2) to explore the relationships between soil respiration rate, soil temperature, soil water content, soil oxygen content, soil bacterial biomass and root biomass under AI and CK conditions, to better understand the links among these soil variables; (3) based on the variable importance for the projection (VIP) method, the main influencing factors of soil respiration rate changes under aerated irrigation were revealed.

## 2. Materials and Methods

### 2.1. Experimental Site

The experiment was conducted from 4 April 2020 to 15 November 2021 in the National Soil Quality Zhanjiang Observation Experiment Station of the Chinese Academy of Tropical Agricultural Sciences in Zhanjiang, Guangdong (E109°31′, N21°35′), with an average annual sunshine duration of 2160 h, a frost-free period of 350 days, and an average annual temperature of 23.2 °C, typical of a subtropical monsoon climate. The test soil was a red loam soil in the original state of the cornfield. The basic physical and chemical properties of soil in the experimental area are shown in Table 1, and the sample sampling date is 26 March 2020. Small weather stations in the study area automatically acquired and recorded rainfall, temperature and other environmental factors during the experiment. The daily

temperature and rainfall variation during the corn growing season in 2020 and 2021 are shown in Figure 1.

**Table 1.** The soil's physical and chemical properties and contents of N, P and K at different depths in the experimental area.

| Soil Depth/(cm) | Soil Bulk Density/ (g·cm⁻³) | The Organic Matter/ (g·kg⁻¹) | N/(mg·kg⁻¹) | P/(mg·kg⁻¹) | K/(mg·kg⁻¹) |
|---|---|---|---|---|---|
| 0–20 | 1.58 | 20.17 | 90.29 | 38.51 | 87.98 |
| 20–40 | 1.61 | 17.16 | 76.12 | 29.66 | 70.17 |
| 40–60 | 1.63 | 15.65 | 65.15 | 33.85 | 76.38 |
| 60–80 | 1.70 | 12.78 | 62.12 | 31.66 | 68.98 |
| 80–100 | 1.73 | 10.42 | 60.09 | 27.78 | 71.97 |

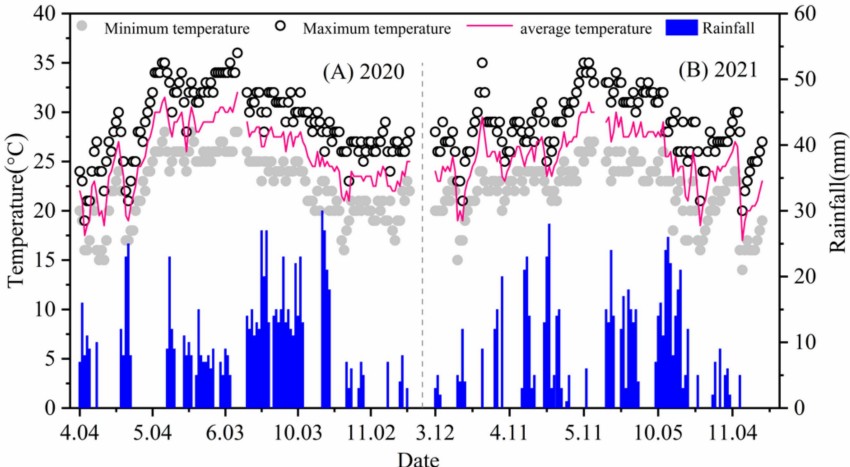

**Figure 1.** Variation curve of average temperature (°C), maximum temperature (mm), minimum temperature (mm) and rainfall in each growth period of corn in Zhanjiang Experimental Station from 2020 to 2021.

### 2.2. Experimental Design and Treatments

The experiment was set up with aerated irrigation (AI) and non-aerated irrigation (CK), and each experiment was replicated three times, with one replication for one plot and a total of six plots. The experimental area was planted with a biannual maize cultivation pattern in the region, and the maize variety planted was "Huiyutian No. 3". Before planting, one underground drip irrigation belt (burial depth 20 cm, diameter 16 mm, flow rate 2.5 L/h, drip head spacing 20 cm) was laid in the middle of each plot. The maize rows were spaced 60 cm apart, and the plants were spaced 35 cm apart (Figure 2).

Irrigation was applied every 2–3 days, between 08:00 and 12:00, based on the total evaporation (measured daily) following the last irrigation event, as determined by an E601 evaporation pan:

$$W = A \cdot E_P \cdot K_P \tag{1}$$

where $W$ (L) is the irrigation amount; $A$ (m²) is the plot area controlled by a single irrigation dripper, which in this experiment is 0.14 m² (0.35 m × 0.4 m); $Ep$ (mm) is the total evaporation following the last irrigation event; and $Kp$ is the crop coefficient for maize. According to previous studies in local areas, the Kc was set to 0.7 from the sowing to the seeding stage, to 1.04 from the jointing to the filling stage, and to 0.6 from the milk to the maturity stage.

The entire reproductive cycle was aerated once every 2 d, with one additional aerating after each irrigation or rainfall, and the amount of aerating was calculated by the Equation (2) [6], without considering the escape of gas from the soil in the experiment.

$$V = 1/1000SL(1 - \rho_b/\rho_s) \tag{2}$$

where, $V$ is the volume per aeration, L; $S$ is the cross-sectional area of the monopoly, 1500 cm$^2$; $L$ is the length of the monopoly, m; $\rho_b$ is the soil capacitance, 1.2 g/cm$^3$; and $\rho_s$ is the soil density, 1.65 g/cm$^3$.

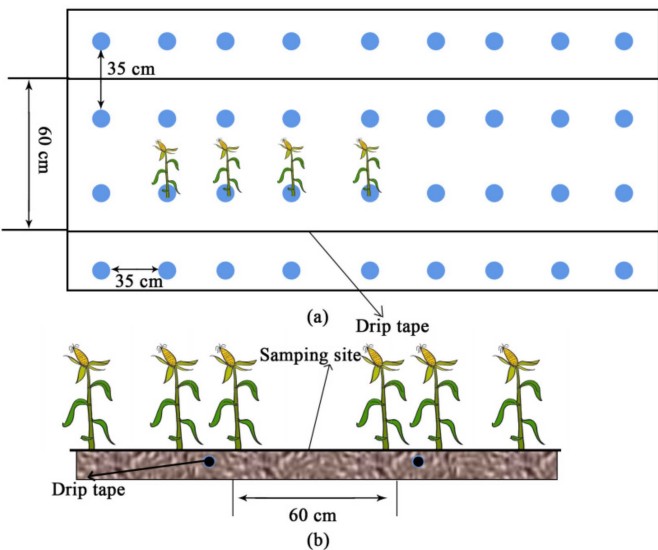

**Figure 2.** Vertical view (**a**) and front view (**b**) of the maize planting patt.

*2.3. Measurement of Soil Respiration, Water Content, Oxygen Content, Bacterial Biomass, Root Biomass and Maize Growth*

The soil respiration rate was monitored with a portable soil $CO_2$ flux system (Li-6400XT, Li-Cor Inc., Lincoln, NE, USA) connected to a Li-6400-09 chamber. Measurements were made during the time period 07:00–09:00, and related studies have shown that the soil respiration rate measured during this time period is representative of the mean soil respiration rate for the day [34]. Diurnal variation in SR was measured at 2-h intervals over a 24-h period starting at 07:00 [35]. In addition to the start date and harvest date, measurements were taken every 10 days during the maize growth cycle, or postponed in case of heavy rainfall, and the average values of different fertility periods of maize were taken for statistical analysis in each trial.

The soil water content and soil temperature were mainly measured by a TZS-PHW-4G soil multi-function parameter tester produced by the Zhejiang Tuopuyun limited company, which measure soil temperature, water content, pH and salinity, and can automatically store data. The technical parameters are shown in Table 2. The measurement depth is 20 cm from the soil, three points are measured each time, and the average value is taken as the measurement result of the group. The measurement date and time coincide with the measurement of soil respiration rate.

**Table 2.** TZS-PHW-4G Multifunctional Soil Parameter Tester.

| Technical Parameter | Unit | Test Range | Precision | Resolution |
|---|---|---|---|---|
| Water content | % | 0–100% | ≤3% | 0.1% |
| pH | - | 0–14 | ±0.5 | 0.1 |

The soil oxygen content was monitored using Robust Oxygen Miniprobes with a Fiber-Optic Oxygen Meter (Firesting $O_2$, PyroScience GmbH, Aachen, Germany). The probe was inserted at a depth of 10 cm, 5 cm from the crop stalk, and the measurement date and time were consistent with the soil respiration rate measurements.

Soil bacterial biomass: Bacteria account for about 94% of the soil microbial composition, and the rest are actinomycetes and fungi, about 4~5% [36]. Therefore, this study chose bacterial biomass as the evaluation index. The soil auger and five-point sampling method

were used to collect the cultivated soil during the measurement and fresh soil samples according to 0~10, >10~20 and >20~30 cm soil layers, and to mix them thoroughly according to the layers. This was repeated 3 times for each collection, using the plate counting method to determine the number of soil bacteria, and measurements were taken approximately every 10 days.

Root biomass: Focus on selected plants, used soil auger to collect the soil layer with a depth of 0~80 cm to screen the root system. The root–soil separation adopts the panning method, the sample is sieved repeatedly after soaking and stirring, after the root soil is separated, the root system is taken out with tweezers, dried and weighed, and then measured approximately every 10 days.

Maize growth traits: Two uniformly growing plants were selected from each plot before harvest to measure their plant height and stalk thickness using tape measure and vernier calipers, respectively, and the leaf area = leaf length × leaf width × 0.75.

Data on yields: After the maize matures, the maize in the test plot is weighed after harvest and converted into a hectare yield according to the planting density.

*2.4. Data Analysis*

All statistical analyses were computed by the SPSS 22.0 software (IBM Corp, Armonk, NY, USA) and SigmaPlot 10.0 (Systat Software, San Jose, CA, USA). An analysis of variance (ANOVA) was used to test the differences in soil respiration rate with soil temperature, water content, oxygen content, soil bacterial biomass and root biomass factors between the two treatments. To quantify the relationship between soil respiration rate and soil temperature, water content, oxygen content, soil bacterial biomass and root biomass, ANOVA and regression analyses were conducted by quadratic, power, exponential and linear models, to filter out the best fit equation through the coefficient $R^2$.

**3. Results**

*3.1. Diurnal Patterns of Soil Respiration, Temperature and Oxygen Concentration*

The soil respiration rate generally showed a unimodal diurnal pattern (Figure 3). The spring–summer trial maximum occurred at 11:00, with the maximum value occurring at 15:00, and the fall–winter trial around 17:00. The soil respiration rate in the two treatments showed similar diurnal patterns, but it was significantly higher with AI than with CK ($p < 0.05$). The mean values of soil respiration rate were 2.92 and 2.66 μmol·m$^{-2}$·s$^{-1}$ in the AI and CK treatments, respectively.

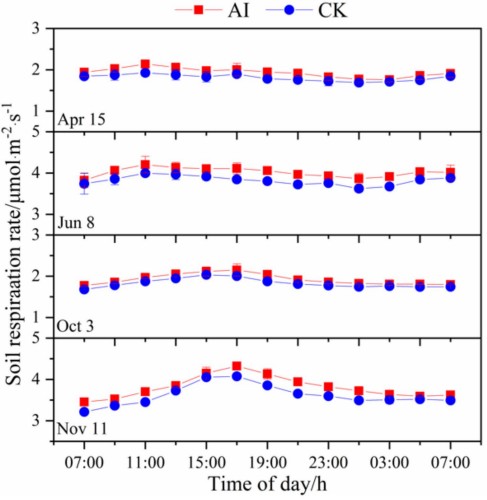

**Figure 3.** Diurnal variation patterns in soil respiration rate based on six measurements under aerated irrigation (AI) and unaerated subsurface drip irrigation (CK) conditions. Diurnal variation was measured at 2-h intervals over a 24-h period starting at 07:00. The bars indicate standard errors of three replications.

The diurnal patterns of soil temperature were similar to those of soil respiration rate with an unimodal pattern (Figure 4), reaching the maximum value at 17:00~19:00 and the minimum value at approximately 01:00. In general, soil temperature was higher in the AI treatment than in the CK treatment, but the difference was not significant under different treatments.

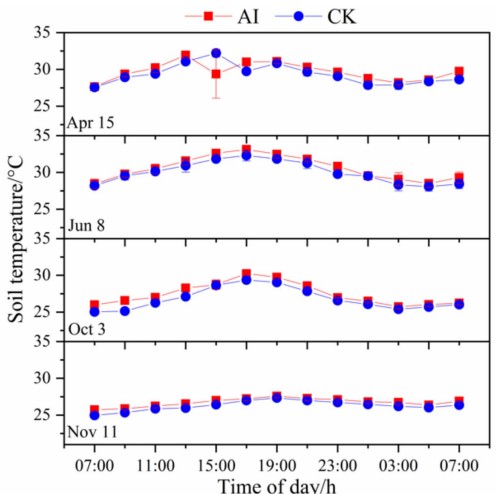

**Figure 4.** Diurnal variation patterns in soil temperature based on six measurements under aerated irrigation (AI) and unaerated subsurface drip irrigation (CK) conditions. Diurnal variation was measured at 2-h intervals over a 24-h period starting at 07:00. The bars indicate standard errors of three replications.

The diurnal patterns of soil oxygen content started with the highest observations at the beginning and end of the measurement period, and minimum values at 15:00–17:00 (Figure 5). The observed soil oxygen content values were consistently higher under the AI treatment than under the CK treatment ($p < 0.01$). The mean values of soil oxygen content under the AI and CK treatments were 14.2% and 12.4%, respectively.

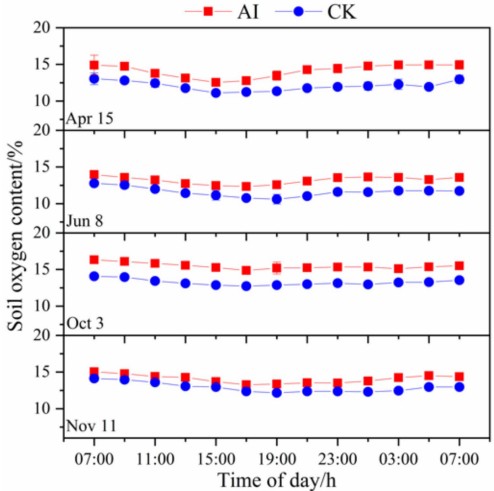

**Figure 5.** Diurnal variation patterns in soil oxygen content under aerated irrigation (AI) and unaerated subsurface drip irrigation (CK) conditions. Diurnal variation was measured at 2-h intervals over a 24-h period starting at 07:00. The bars indicate standard errors of three replications.

### 3.2. Seasonal Variation in Various Soil Parameters Due to Aerated Irrigation

3.2.1. Soil Respiration Rate

There was some variability in the seasonal dynamics of soil respiration rate (in terms of $CO_2$) under different treatments, and the measured points were all significantly higher

under AI treatment than the control CK treatment ($p < 0.05$)(Figure 6). The statistics of soil respiration rate changes under different treatments are shown in Table 3. In the spring–summer crop trials from 2020 to 2021, the mean values of soil respiration rate changes under AI and CK treatments were 3.07 μmol·m$^{-2}$·s$^{-1}$, 3.10 μmol·m$^{-2}$·s$^{-1}$ and 2.62 μmol·m$^{-2}$·s$^{-1}$, 2.63 μmol·m$^{-2}$·s$^{-1}$, respectively, and the seasonal changes in soil respiration were significantly greater by 17.18% and 17.87% for AI compared with the control CK ($p < 0.05$).The mean values of soil respiration rate changes under AI and CK treatments from 2020 to 2021 fall–winter crop trials were 2.89 μmol·m$^{-2}$·s$^{-1}$, 2.55 μmol·m$^{-2}$·s$^{-1}$ and 2.48 μmol·m$^{-2}$·s$^{-1}$, 2.21 μmol·m$^{-2}$·s$^{-1}$, respectively, and the seasonal changes in soil respiration under AI were significantly greater than the control CK by 16.53% and 15.38% ($p < 0.05$).

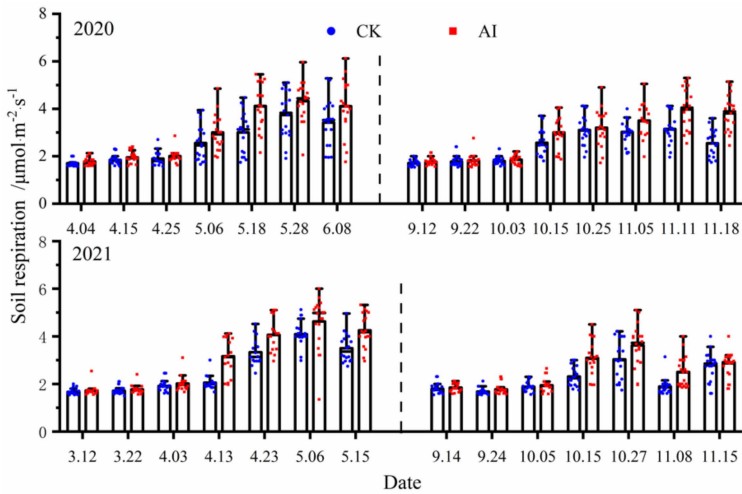

**Figure 6.** The dynamic change curve of soil respiration rate under different treatments (the dots represent the actual measured value of the repeated test, and the same below).

**Table 3.** Change characteristics of soil respiration rate and environmental factors under different treatments.

| Indicators/ Year | | Rs/μmol m$^{-2}$·s$^{-1}$ | | Ts/°C | | Ws/% | |
|---|---|---|---|---|---|---|---|
| | | AI | CK | AI | CK | AI | CK |
| Spring/ Summer 2020 | Maximum | 6.12 | 5.28 | 32.0 | 31.4 | 55.1 | 58.6 |
| | Minimum | 1.59 | 1.57 | 23.0 | 22.2 | 15.3 | 14.6 |
| | Mean | 3.07 ± 1.28 | 2.62 ± 1.03 | 28.11 ± 3.44 | 27.66 ± 3.22 | 28.41 ± 10.95 | 30.47 ± 11.14 |
| Fall/ Winter 2020 | Maximum | 5.3 | 4.12 | 31.9 | 31.0 | 54.1 | 56.0 |
| | Minimum | 1.54 | 1.54 | 26.2 | 25.8 | 14.0 | 16.0 |
| | Mean | 2.89 ± 1.05 | 2.48 ± 0.74 | 28.83 ± 1.42 | 28.50 ± 1.28 | 24.32 ± 8.80 | 27.14 ± 9.93 |
| Spring/ Summer 2021 | Maximum | 5.12 | 5.12 | 31.5 | 31.8 | 45.7 | 48.7 |
| | Minimum | 1.5 | 1.35 | 22.7 | 22.3 | 16.0 | 16.0 |
| | Mean | 3.10 ± 1.02 | 2.63 ± 1.34 | 27.81 ± 2.46 | 27.64 ± 2.63 | 25.16 ± 6.59 | 26.50 ± 7.73 |
| Fall/ Winter 2021 | Maximum | 5.1 | 4.21 | 30.6 | 30.6 | 58.7 | 59.4 |
| | Minimum | 1.58 | 1.54 | 20.6 | 19.6 | 15.6 | 16.4 |
| | Mean | 2.55 ± 0.85 | 2.21 ± 0.66 | 27.38 ± 3.16 | 27.14 ± 3.23 | 30.82 ± 8.63 | 32.62 ± 9.82 |
| Indicators/ Year | | Os/% | | Bs/10$^9$·g$^{-1}$ | | Rb/g | |
| | | AI | CK | AI | CK | AI | CK |
| Spring/ Summer 2020 | Maximum | 16.6 | 16.4 | 6.0 | 4.5 | 36.85 | 30.59 |
| | Minimum | 10.8 | 10.0 | 1.0 | 0.5 | 0.00 | 0.00 |
| | Mean | 14.43 ± 1.33 | 12.01 ± 1.24 | 3.80 ± 1.52 | 2.61 ± 1.44 | 17.87 ± 15.28 | 15.34 ± 13.42 |
| Fall/ Winter 2020 | Maximum | 18.2 | 15.6 | 6.0 | 5.0 | 36.80 | 31.20 |
| | Minimum | 11.0 | 10.4 | 1.5 | 0.5 | 0.00 | 0.00 |
| | Mean | 14.67 ± 1.56 | 11.72 ± 1.12 | 3.99 ± 1.31 | 2.89 ± 1.08 | 20.17 ± 15.01 | 17.54 ± 13.16 |
| Spring/ Summer 2021 | Maximum | 17.5 | 15.4 | 5.5 | 5.0 | 37.32 | 33.40 |
| | Minimum | 11.0 | 10.5 | 1.0 | 0.5 | 0.00 | 0.00 |
| | Mean | 14.88 ± 1.58 | 12.51 ± 1.26 | 4.08 ± 1.35 | 3.02 ± 1.22 | 17.71 ± 15.28 | 15.03 ± 13.38 |
| Fall/ Winter 2021 | Maximum | 17.2 | 16.4 | 6.5 | 4.5 | 36.45 | 31.41 |
| | Minimum | 11.0 | 9.5 | 1.0 | 0.5 | 0.00 | 0.00 |
| | Mean | 14.52 ± 1.49 | 12.14 ± 1.49 | 4.06 ± 1.53 | 2.83 ± 1.06 | 18.53 ± 15.35 | 15.56 ± 13.35 |

Note: AI, aerated irrigation; CK, unaerated subsurface drip irrigation; Rs, soil respiration; Ts, soil temperature; Ws, soil water content; Os, soil oxygen content; Bs, soil bacterial biomass; Rb, root biomass.

### 3.2.2. Soil Temperature

There was basically no significant difference in all measurement points of soil temperature under different treatments ($p < 0.05$). Soil temperature under the AI treatment was higher than that under the CK treatment in only some of the measured points of seasonal variation (Figure 7). The statistics of soil temperature changes under different treatments are shown in Table 3. Soil temperature changes are closely related to climate change. In the spring–summer crop trials from 2020 to 2021, the mean values of soil temperature changes under the AI and CK treatments were 28.11 °C, 27.81 °C and 27.81 °C, 27.64 °C. The mean values of soil temperature changes under AI and CK treatments from 2020 to 2021 fall–winter crop trials were 28.83 °C, 27.38 °C and 28.50 °C, 27.14 °C. Soil temperature was the main factor influencing the variation of soil respiration rate, which was also evident from the fact that the peak soil temperature clearly corresponded to the peak soil respiration rate (Figures 6 and 7).

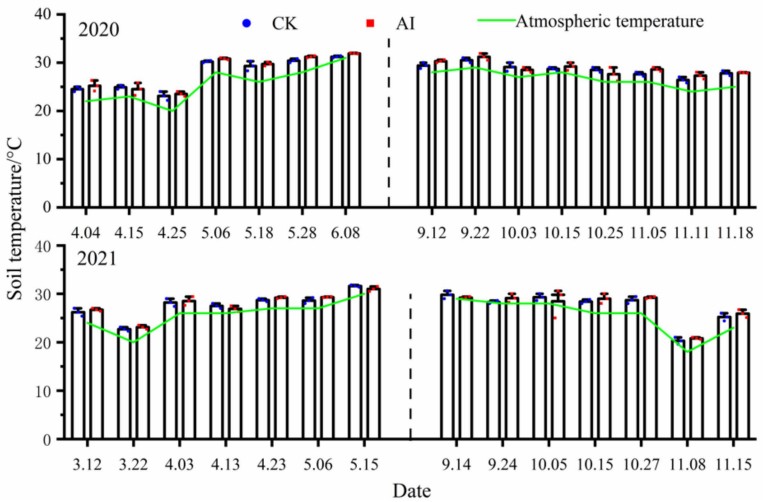

**Figure 7.** The dynamic change curve of soil temperature under different treatments.

### 3.2.3. Soil Oxygen Content

In this study, all the soil oxygen measurement points under AI treatment were significantly higher than the control treatment ($p < 0.05$) (Figure 8). The statistics of soil oxygen content changes under different treatments are shown in Table 3. In the spring–summer crop trials from 2020 to 2021, the mean values of soil oxygen content changes under the AI and CK treatments were 14.43%, 14.88% and 12.01%, 12.51%, respectively, and the seasonal changes in soil oxygen content were significantly greater by 20.15% and 18.94% for AI compared with the control CK ($p < 0.05$). The mean values of soil oxygen content changes under the AI and CK treatments from 2020 to 2021 fall–winter crop trials were 14.67%, 14.52% and 11.72%, 12.14%, respectively, and the seasonal changes in soil oxygen content under AI were significantly greater than the control CK by 25.17% and 19.60% ($p < 0.05$).

### 3.2.4. Soil Water Content

Soil water content in this study fluctuated widely and was mainly affected by rainfall and irrigation, with rainfall mainly concentrated from September to October. Except for the sudden increase in soil water content due to rainfall, the highest water content was found in all experimental treatments at the maize seedling stage (Figure 9), mainly due to the watering of the bottom water before maize planting. The soil water content measurement points under the AI treatment were basically lower than those of the control test, and the statistics of soil water content change characteristics under different treatments in the 2-year trial are shown in Table 3. In the spring–summer crop trials from 2020 to 2021, the mean values of soil water content changes under the AI and CK treatments were 28.41%, 25.16% and 30.47%, 26.50%, respectively, and the seasonal variation of soil water content under the

AI treatment decreased by 7.25% and 5.33%, compared to the control CK ($p < 0.05$). The mean values of soil water content changes under the AI and CK treatments from 2020 to 2021 fall–winter crop trials were 24.32%, 30.82% and 27.41%, 32.62%, respectively, and the seasonal variation of soil water content under AI treatment decreased by 12.71% and 5.84% compared to the control CK ($p < 0.05$).

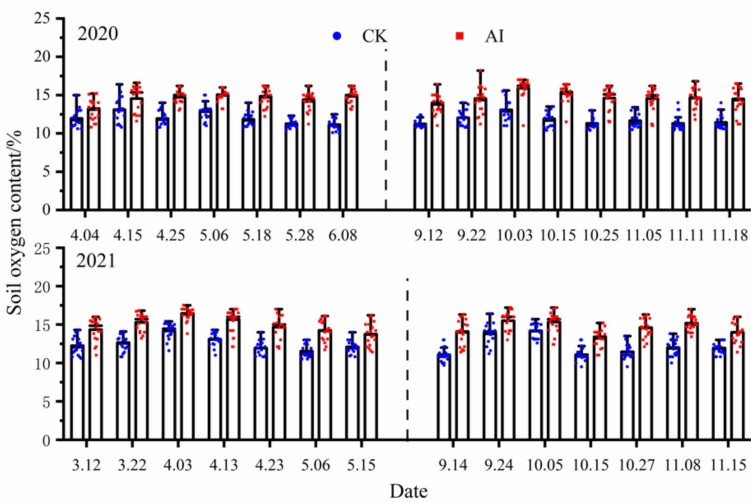

**Figure 8.** The dynamic change curve of soil oxygen content under different treatments.

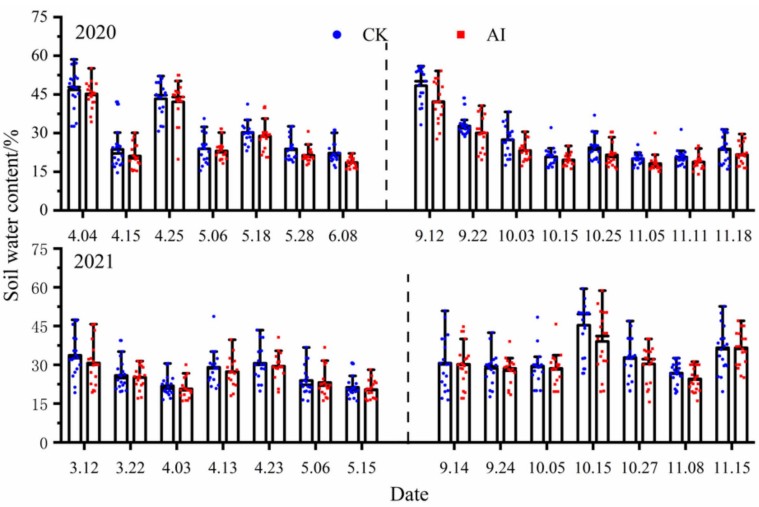

**Figure 9.** The dynamic change curve of soil water content under different treatments.

### 3.2.5. Soil Bacterial Biomass

Soil bacteria accounted for about 70–90% of the total soil microorganisms and were the main influencing factor on soil respiration. There were basically significant differences ($p < 0.05$) in all measurement points of soil bacterial biomass under different treatments (Figure 10), and the statistics of soil bacterial biomass change characteristics under different treatments in the 2-year experiment are shown in Table 3. In the spring–summer crop trials from 2020 to 2021, the mean values of soil bacterial biomass changes under the AI and CK treatments were $3.80 \times 10^9 \cdot g^{-1}$, $4.08 \times 10^9 \cdot g^{-1}$ and $2.61 \times 10^9 \cdot g^{-1}$, $3.02 \times 10^9 \cdot g^{-1}$, respectively, and the seasonal changes in soil bacterial biomass were significantly greater by 45.59% and 35.10% for AI compared with the control CK ($p < 0.05$). The mean values of soil bacterial biomass changes under the AI and CK treatments from 2020 to 2021 fall/winter crop trials were $3.99 \times 10^9 \cdot g^{-1}$, $4.06 \times 10^9 \cdot g^{-1}$ and $2.89 \times 10^9 \cdot g^{-1}$, $2.83 \times 10^9 \cdot g^{-1}$, respectively, and the seasonal changes in soil bacterial biomass under AI were significantly greater than the control CK by 38.06% and 43.46% ($p < 0.05$).

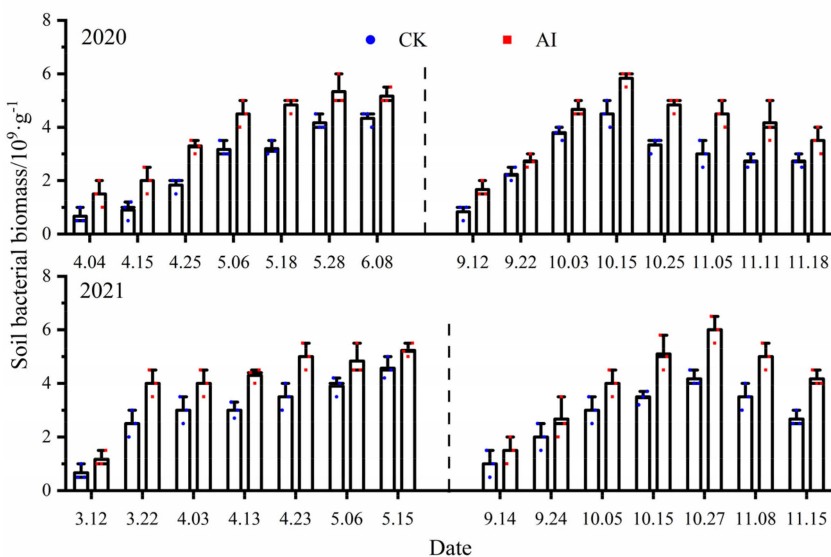

**Figure 10.** The dynamic change curve of soil bacterial biomass under different treatments.

### 3.2.6. Root Biomass

The root biomass under different treatments showed three growth stages of "first slow increase, second rapid increase, and finally slow increase" with time after planting (Figure 11). At the seedling stage, root biomass tended to increase slowly, and the increase was more consistent under different treatments. After maize entered the jointing stage, root growth and development were faster, and after entering the maturity stage, root biomass showed a small increase, and later there was a tendency to decrease, but the range of change was small. The root biomass under AI treatment was higher than the control under different experiments, and the statistics of root biomass variation characteristics under different treatments are shown in Table 3. In the spring–summer crop trials from 2020 to 2021, the mean values of root biomass changes under AI and CK treatments were 17.87 g, 17.71 g and 15.34 g, 15.03 g, respectively. The seasonal variation of root biomass under the AI treatment decreased by 16.49% and 17.83% compared to the control CK ($p < 0.05$). The mean values of root biomass changes under the AI and CK treatments from 2020 to 2021 fall–winter crop trials were 20.17 g, 18.53 g and 17.54 g, 15.56 g, respectively, and the seasonal variation of root biomass under AI treatment decreased by 14.99% and 19.09% compared to the control CK ($p < 0.05$).

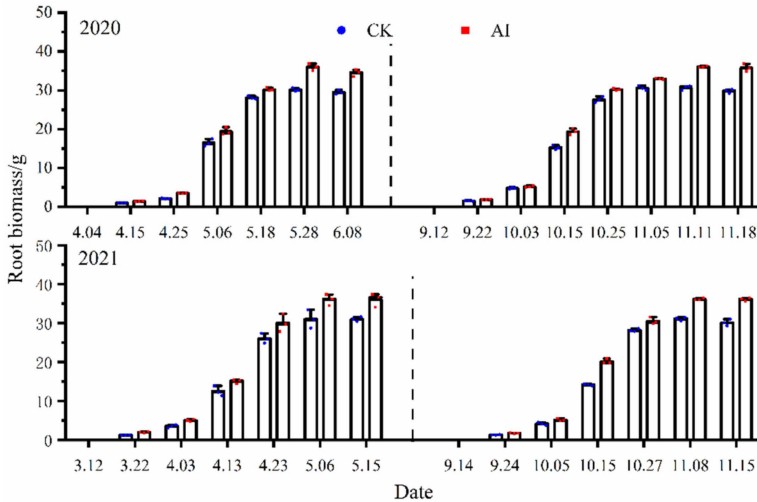

**Figure 11.** Changes of maize root biomass factors under different treatments.

### 3.3. Effects of Aerated Irrigation on the Growth of Maize

3.3.1. Plant Height, Stem Thickness and Leaf Area

Aerated irrigation had no significant effect on maize plant height (Table 4), but there was a highly significant positive response of stem thickness and leaf area to aerated irrigation compared to the control ($p < 0.01$). Maize stem thickness significantly increased by 9.31% to 17.06%, and the leaf area increased significantly, by 8.68% to 15.20%, under aerated irrigation.

**Table 4.** Plant height, stem thickness and leaf area under different treatments.

| Year | | Plant Height/cm | | Stem Thickness/mm | | Leaf Area/cm² | |
|---|---|---|---|---|---|---|---|
| | | AI | CK | AI | CK | AI | CK |
| Spring/ Summer 2020 | Maximum | 231 | 221 | 34.9 | 30.1 | 4910.71 | 4521.63 |
| | Minimum | 182 | 191 | 28.9 | 26.5 | 4491.02 | 4012.31 |
| | Mean | 216.43 ± 18.72 | 201.43 ± 11.10 | 31.01 ± 2.10 | 28.37 ± 1.56 | 4778.65 ± 166.13 | 4300.55 ± 209.36 |
| Difference analysis | | No | | * | | ** | |
| Fall/ Winter 2020 | Maximum | 236 | 248 | 32.1 | 30.5 | 4967.23 | 4411.21 |
| | Minimum | 198 | 184 | 28.7 | 21.9 | 4413.62 | 3987.45 |
| | Mean | 218.33 ± 15.04 | 220 ± 21.00 | 30.32 ± 1.19 | 27.17 ± 2.96 | 4656.65 ± 212.51 | 4208.43 ± 157.80 |
| Difference analysis | | No | | * | | ** | |
| Spring/ Summer 2020 | Maximum | 250 | 232 | 32.4 | 31 | 4975.06 | 4413.68 |
| | Minimum | 182 | 188 | 27.8 | 22.6 | 4432.56 | 3874.25 |
| | Mean | 219.33 ± 25.30 | 212.67 ± 14.28 | 29.97 ± 1.79 | 27.13 ± 3.08 | 4807.84 ± 191.83 | 4173.38 ± 177.20 |
| Difference analysis | | No | | No | | ** | |
| Fall/ Winter 2020 | Maximum | 243 | 245 | 31.5 | 28.5 | 4971.23 | 4521.09 |
| | Minimum | 200 | 186 | 28.2 | 23.3 | 4219.64 | 4025.33 |
| | Mean | 221.17 ± 15.22 | 212 ± 21.82 | 29.58 ± 1.24 | 25.27 ± 2.18 | 4600.92 ± 271.35 | 4233.65 ± 1989.93 |
| Difference analysis | | No | | ** | | * | |

Note: AI: aerated irrigation; CK: unaerated subsurface drip irrigation. No, No significant difference; *, significance at the 0.05 level; **, significance at the 0.01 level.

3.3.2. Maize Yield

Compared to that with the CK treatment, the maize yield and per maize weight significantly increased by 7.16–20.51% and −1.58–1.49%, respectively, with the AI treatment ($p < 0.05$) (Table 5).

**Table 5.** Maize yield and fruit weight under different treatments.

| Year | | Maize Yield (kg/ha) | | Weight Per Fruit(g) | |
|---|---|---|---|---|---|
| | | AI | CK | AI | CK |
| Spring/ Summer 2020 | Maximum | 22,400 | 21,450 | 300.5 | 312.5 |
| | Minimum | 20,654 | 17,750 | 292.6 | 294.6 |
| | Mean | 21,359.67 ± 919.85 | 19,933.33 ± 1938.00 | 297.77 ± 4.48 | 302.4 ± 9.17 |
| Difference analysis | | * | | * | |
| Fall/ Winter 2020 | Maximum | 24,069 | 22,300 | 324.9 | 324.5 |
| | Minimum | 18,090 | 15,600 | 298.2 | 300.1 |
| | Mean | 21,391 ± 3037.80 | 19,366.67 ± 3426.85 | 312.6 ± 13.47 | 308.27 ± 14.06 |
| Difference analysis | | ** | | No | |
| Spring/ Summer 2021 | Maximum | 29,202 | 24,600 | 300.5 | 312.5 |
| | Minimum | 21,111 | 18,500 | 292.6 | 294.6 |
| | Mean | 25,604.33 ± 4119.19 | 21,246.67 ± 3094.92 | 297.77 ± 4.48 | 302.4 ± 9.17 |
| Difference analysis | | * | | No | |
| Fall/ Winter 2021 | Maximum | 23,069 | 21,200 | 324.9 | 324.5 |
| | Minimum | 19,014 | 17,230 | 298.2 | 300.1 |
| | Mean | 21,057.67 ± 2027.69 | 18,663.33 ± 2203.01 | 312.6 ± 13.47 | 308.27 ± 14.06 |
| Difference analysis | | ** | | No | |

Note: AI: aerated irrigation; CK: unaerated subsurface drip irrigation. No, No significant difference; *, significance at the 0.05 level; **, significance at the 0.01 level.

### 3.4. Correlation Analysis between Soil Respiration Rate and Soil Environmental Factors under Different Treatments

The test results were fitted and analyzed (excluding the abnormal data in the test), and the relationship between soil respiration rate and each influencing factor was fitted by segmentation using a linear model, nonlinear model and polynomial model, and the best-fit equation was screened by the coefficient $R^2$ (Figure 12). Soil temperature was an important environmental factor affecting soil respiration rate, and there was a negative quadratic polynomial correlation ($p < 0.05$) between soil temperature and soil respiration rate under both experimental treatments (Figure 12a).

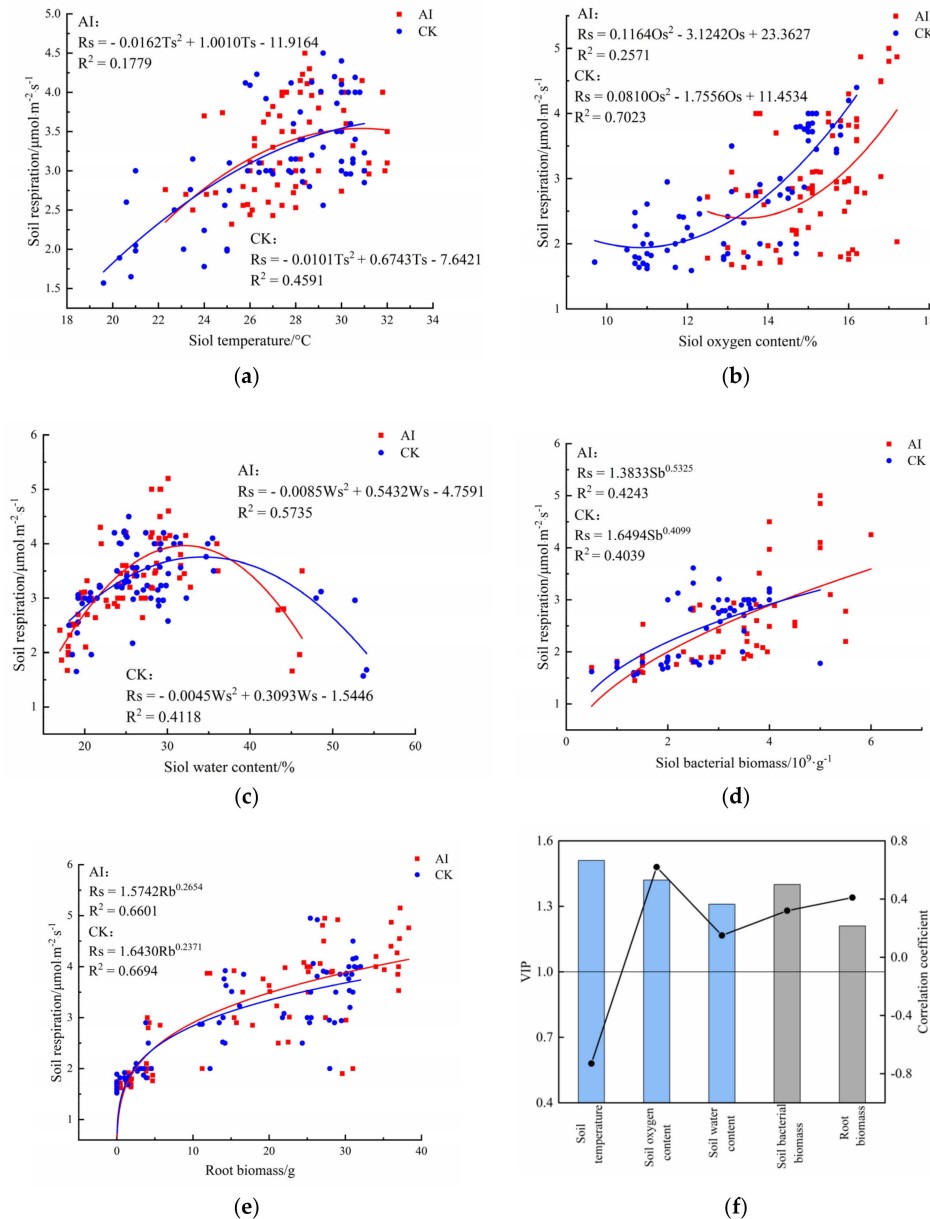

**Figure 12.** Relationships between soil respiration rate and soil temperature (**a**), soil oxygen content (**b**), soil water content (**c**), soil bacterial biomass (**d**) and root biomass (**e**) under AI and CK conditions and the projection of importance of variables affecting soil respiration rate (VIP) and regression coefficients (black line) (**f**). AI, aerated irrigation; CK, control group; The scatter plots show the measured values, and the curves represent the predicted values using the provided fitted equations.

Oxygen content was quadratic polynomial positive correlated with soil respiration rate ($p < 0.05$) (Figure 12b), soil oxygen content varied from 12.5% to 17.2% and 9.7% to

16.1% under the AI and CK treatments, and soil respiration rate peaked at 16.8% under the AI treatment, followed by 16.2% under the CK treatment. Under different treatments, soil respiration rate was enhanced gradually with the increase of soil oxygen content.

There was a negative quadratic polynomial correlation ($p < 0.05$) between water content and soil respiration rate (Figure 12c), and the range of water content variation under the AI and CK treatments was 17.0~46.3% and 18.1~54.1%. The soil respiration rate gradually increased with the increase of water content, and when the soil water content exceeded 32.1%, the soil respiration rate roughly showed a decreasing trend, under CK treatment. When the soil water content ranged from 18.1% to 31.5%, the soil respiration rate gradually increased with the increase of water content, and when the soil water content exceeded 31.5%, the soil respiration rate showed an overall decreasing trend.

There was a positive power function correlation ($p < 0.05$) between soil respiration rate and root biomass. Soil respiration rate was power function positively correlated with root biomass and bacterial biomass under the two treatments. (Figure 12d,e), with root biomass varying from 0 to 37.32 g and 0 to 32.00 g under the AI and CK treatments. The bacterial biomass varied from 0.5 to $6 \times 10^9$ $g^{-1}$ and 0.5 to $5 \times 10^9$ $g^{-1}$ under the AI and CK treatments. The soil respiration rate gradually increased with the increase of root biomass and bacterial biomass under different treatments.

The correlation analysis of soil respiration rate with the main environmental factors is shown in Figure 12f. Soil respiration rate was highly significantly and positively correlated with soil oxygen content and root biomass, with correlation coefficients of 0.62 and 0.32, respectively; it was highly and negatively correlated with soil temperature, with a correlation coefficient of −0.73, and was weakly correlated with soil water content, with a correlation of −0.27. The projection of importance of PLSR variables (VIP) showed that the order of influence of each influencing factor on soil respiration was: soil temperature (VIP = 1.51) > soil oxygen content (VIP = 1.42) > root biomass (VIP = 1.40) > soil water content (VIP = 1.31) > bacterial biomass (VIP = 1.21) (Figure 12f). The correlation analysis showed that the changes in soil oxygen content, root biomass and bacterial biomass under aerated irrigation were the most critical influencing factors on soil respiration rate.

## 4. Discussion

### 4.1. Variation in Various Soil Parameters and Fruit Growth

In this experiment, soil respiration rate and soil oxygen content significantly increased with AI treatment (Figures 3, 5, 6 and 8). The study by Bhattarai [17] showed that the dissolved oxygen with AI was 12% greater than that without aeration. Similarly, the results of this experiment showed that the mean soil oxygen content increased by 18.94~25.17% with AI. Bhattarai [17] also found that the lowest point of soil oxygen content for one diurnal measurement occurred between 14:00 and 16:00, which is slightly different from the results of this experiment, in which the lowest point occurred between 23:00 and 01:00 (Figure 5). Other research by Bhattarai et al. [37] showed that, compared to soil respiration rate with no aeration, soil respiration rate with AI increased by 124 and 183% for zucchini and cotton, respectively. Therefore, the higher soil respiration rate that can be obtained under aerated irrigation treatment is due to the increase in soil oxygen content, which provides a good aerobic environment for plant roots and microbial life metabolic activities, improves root growth and microbial metabolic activity, and thus enhances soil respiration rate. On the other hand, aerating the soil can promote gas exchanges between the soil and the atmosphere [10], which can ensure a smooth soil respiration. The results of Hou, Zhu Y, Zang M and LI Y [21,22,30,38] also showed that the soil respiration rate increased to about 12.5% to 20.1%, respectively, compared to conventional subsurface irrigation.

The soil is a dynamic, three-phase (solid, liquid and gas) system, and the solid phase is relatively stable. Therefore, an increase in water or air inevitably leads to a decrease in the other phases. Effective soil aeration ensures sufficient oxygen diffusion into the root zone for optimal crop functions [39]. Ben-Noah and Friedman demonstrated [24] that injecting air through SDI pushed soil water downward and lowered the soil moisture below the

dripper. Through AI, the decline in soil $O_2$ and the concentration of soil moisture around the plant roots were apparently ameliorated, which is advantageous for enhancing the availability of $O_2$ and gas exchange. Weltecke and Gaertig [40] investigated the aeration and respiration of soils with different soil mulch coverages, and their results showed that SR and gas diffusivity at sealed sites under asphalt, flagstone or cobblestone mulch were significantly (even 10 times) lower than the values at sites without mulch. The soil respiration and gas diffusivity at sealed sites was lower because the mulch disrupted the only connection between the roots, soil and atmosphere-continuous air-filled pores. Hence, with AI, the aeration of soil around the root zone was effectively improved, and the negative effects on soil respiration could be offset. Consequently, in our experiment, the favorable soil aeration conditions (higher soil oxygen content) under AI significantly increased the soil respiration rate. Crop growth and yields largely depend on the balance between soil air, water and nutrients, and if any of these factors reach their upper limit, crop yields will be damaged [41]. With AI, the soil air/water ratio may be closer (than with the CK treatment) to the optimum "fertile triangle" balance [42] between air, water and nutrients. Thus, in the present research, the significantly increased fruit yields obtained with AI might have been due to the ameliorated oxygen deficiency and the appropriate soil air/water ratio. The yield improvements observed in this study were consistent with those in previous research [17,19–21,37,39] on AI, which reveals that AI unlocks a crop yield potential under various developmental stages.

Crop root growth was more sensitive to low oxygen stress, and root biomass increased by 14.99~19.09% under aerated irrigation compared with the control group, mainly because the improvement of soil environment provided a good growing environment for crop root growth, met the root demand for soil oxygen, enhanced the root uptake of soil water and nutrients, and promoted crop root growth and development. Under different treatments, soil root biomass showed a slow growth from the seedling stage to the nodulation stage, with slow root growth during the seedling stage and rapid root growth during the jointing stage to the filling stage, reaching a maximum after the maize milk stage and then leveling off (Figure 11). Aerated irrigation increased microbial activity based on improved soil aeration, the AI treatment could significantly increase soil bacterial biomass by 35.10% to 45.59% compared to the control, and the effect was more significant as the year increased (Figure 10). The seasonal variation trends of soil bacterial biomass were similar to those of soil respiration, that is, in the spring and summer stubble experiments, bacterial biomass generally showed an increasing trend, while in the autumn and winter stubble experiments, they generally showed a seasonal trend of first increasing and then decreasing (Figure 10).

*4.2. Relationships of Soil Respiration with Water Content, Oxygen Content, Bacterial Biomass and Root Biomass*

Soil temperature, water content and oxygen content are the main soil environmental factors that influence changes in soil respiration rate. Soil temperature is closely related to the rate of soil respiration and affects almost all aspects of soil respiration, including humus decomposition, root growth and the conduct of all life activities of soil microorganisms [43]. Liu et al. suggested that temperature variation could explain most of the variation in soil respiration rate changes [44]. The regression fitting of soil temperature and soil respiration rate in this study showed that soil temperature was negatively correlated with soil respiration rate by a quadratic polynomial (Figure 12a), and soil respiration rate began to gradually decrease when soil temperature reached 31.2 °C and 30.6 °C. This was inconsistent with the studies of Zhu Y [21], which was mainly due to the different test locations and test climates. A related study by Arredondo et al. showed that a high temperature can have some inhibitory effect on soil respiration rate [45].

In this study, the soil oxygen and soil respiration is positively correlated ($p < 0.05$) under the different treatments (Figure 12b). Arredondo, Hursh and Anna [45–47] showed that soil respiration and soil gas diffusion rate are positively correlated, as the soil oxygen increases with the increase of soil gas diffusion rate. Soil water content is also the main

controlling factor affecting soil respiration, and through the fitting analysis, soil water content showed a negative quadratic polynomial correlation with soil respiration rate (Figure 12c), and soil respiration rate did not change significantly with soil water content when soil water content was between 24.5% and 35.2%. HOU Maomao et al. [48] showed that only when soil water content was below or above the threshold value was the soil respiration rate significantly correlated with soil water content, and when soil water content was between 24.5% and 36.7%, the correlation with soil respiration rate was weak. In this study, soil water content was mainly concentrated in 24.5% to 35.2%, and when soil water content was lower than 20.4% and higher than 40.2%, it caused obvious changes to the trend of soil respiration (Figure 12c), which was due to the fact that soil respiration could not diffuse to the atmosphere because soil moisture increased, aeration in the soil became poor, soil oxygen content decreased, soil respiration was inhibited and $CO_2$ suddenly dropped.

Soil respiration is not only influenced by soil abiotic factors, but more often by biotic factors, and the number of plant roots and microorganisms in the soil is an important biotic factor affecting soil respiration rate. Soil microbial respiration accounts for 40% of soil respiration, root respiration accounts for 40% to 50% of soil respiration, and the rest is $CO_2$ production from the decomposition of apoplastic and organic matter in the soil. It has been shown that an increase in soil microbial load enhances soil respiration, and the soil respiration rate is gradually enhanced with an increase in soil bacterial biomass and root biomass [49]. In this study, root biomass and bacterial biomass were positively correlated with soil respiration rate as a power function (Figure 12d,e), which is consistent with the findings of previous studies.

The projection of variable importance (VIP) shows that the main order of influence of each influence factor on soil respiration was: soil temperature (VIP = 1.51) > soil oxygen content (VIP = 1.42) > root biomass (VIP = 1.40) > soil water content (VIP = 1.31) > bacterial biomass (VIP = 1.21) (Figure 12f). Soil bacteria and root biomass play a crucial role in soil carbon release [50]. However, there may be significant redundancy in soil microbial and plant root functions, and changes in soil biological properties below a certain threshold may have little effect on soil function [51]. Thus, soil microbial and plant root mineralization is strongly regulated by soil abiotic factors such as temperature, oxygen content, pH and active organic matter content, which is consistent with the results of this study. Although aerated irrigation had some effect on raising soil temperature, the effect was not significant and the change in soil temperature was mainly related to atmospheric temperature (Figure 7) and not much related to whether aerated treatment was applied or not. Therefore, it can be concluded that soil oxygen content, root biomass and bacterial biomass are the most important environmental and biological factors affecting soil respiration rate under the effect of aerated irrigation.

*4.3. Recommendation and Limitations*

Previous researches on AI often focused on its apparent impacts, such as on plant height, stem diameter, leaf area, crop biomass, water use efficiencies and crop yields. However, the most essential improvement in SDI due to AI is the amelioration of the soil micro-environment due to optimization of the soil's air/water ratio. This study not only analyzed the impacts of AI on soil respiration rate, soil temperature, water content, oxygen content, soil bacterial biomass and root biomass, but also explored the relationships between these parameters. However, our study only examined a subset of the factors influencing the soil micro-environment, and further research on AI should systematically analyze the impacts of AI on other soil parameters, such as soil enzymes, soil gas (including $CH_4$ and $N_2O$) exchanges and fine root growth parameters around the root zone. Further research in this field will be carried out in the future.

**5. Conclusions**

From a pilot study for two consecutive years, from 2020 to 2021, soil respiration rate showed a trend of increasing and then decreasing at different fertility stages of maize under

different treatments. Compared with the control CK, the AI treatment could significantly increase soil respiration rate, by 15.38% to 17.87%, increase soil oxygen content, root biomass and bacterial biomass, by 19.60% to 25.17%, 14.99% to 19.09% and 35.10% to 45.59%, respectively, and reduce soil water content by 5.33% to 11.60%, and the different treatments on soil temperature were not significantly affected. Consequently, the maize yield with AI was 7.16~20.51% higher than with CK due to the favorable soil conditions, to obtain greater economic benefits.

The correlation analysis showed that under the AI and CK treatments, soil temperature, soil water content and soil oxygen content were negatively correlated with soil respiration rate by a quadratic polynomial, root biomass and soil bacterial biomass were positively correlated with soil respiration rate by a power function, and soil respiration rate gradually increased with root biomass and bacterial biomass.

The projection of variable importance (VIP) showed that the sensitivity of soil respiration rate to each influencing factor differed, and the main order of influence of each influencing factor on soil respiration was: soil temperature (VIP = 1.51) > soil oxygen content (VIP = 1.42) > root biomass (VIP = 1.40) > soil water content (VIP = 1.31) > bacterial biomass (VIP = 1.21). From the above analysis, it is clear that aerated irrigation technology affects soil respiration rate mainly by increasing soil oxygen content and root biomass.

**Author Contributions:** Z.Y., C.W. and H.W. wrote the paper, C.W. designed the study idea and applied themodel to be calculated, Z.Y., D.Y. and H.Z. collected and arranged the all database, H.L. and H.S. analyzed and discussed. All the authors revised the manuscript. All authors have read and agreed to the published version of the manuscript.

**Funding:** This research was funded by 2020 Rural Science and Technology Specialists in Guangdong Province Key Assignment Tasks (Guangdong Science Letter Agricultural Word [2020] 409); and Central Public-Interest Scientific Institution Basal Research Fund for Chinese Academy of Tropical Agricultural Sciences No. 19CXTD-31.

**Institutional Review Board Statement:** Not applicable.

**Informed Consent Statement:** Not applicable.

**Data Availability Statement:** Not applicable.

**Acknowledgments:** We thank Huiyuan Wang, Guoqing Ma, Haochen Zhan and Qichang Dong for providing us with experimental field plots and helping us to collect the data.

**Conflicts of Interest:** The authors declare no conflict of interest.

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
