# Peer review of "The Effects of Aerated Irrigation on Soil Respiration and the Yield of the Maize Root Zone"

_sustainability, doi:10.3390/su14084378_

Round 1
Reviewer 1 Report
The theme addressed in this manuscript: aerated irrigation technology, improves soil aeration conditions and, consequently, improves crop yield, suggesting economic benefits. This work is consistent with the theme of the magazine. It does not have great originality, but it could be very interesting for the readers.
It is also noteworthy that the bibliography used is current, since half of the references are from the last 5 years.
Author Response
Dear reviewers:
Thank you for your letter and for the reviewers'comments concerning our manuscript entitled “The effects of aerated irrigation on soil respiration and yield of maize root zone”(ID:1631071).Those comments are all valuable and very helpful for revising and improving our paper,as well as the important guiding significance to our research.We have studied comments carefully and have made correction which we hope meet with approval.Please see attachment.

Reviewer 2 Report
Abstract: OK
Introduction: L 39-48 not clear, need restructure the sentences.
L 50-51 Pl confirm AI abbreviation for Aerated irrigation and Spell out SDI first
L65-72 Pl mention research gap.
Pl add updated literature
Objectives: excellent
Materials and Methods: Over all ok (for specific comments pl see)
Experimental site: add details of climatic conditions, which is being presented in graph.
Figure 1: quality ok, in X axis add year
Table 1: Add sampling year for clarification
Experimental design and treatments L 101-102 pl cross check both abbreviation
Figure 2: Quality improvement require
Measurement of soil respiration and other: Excellent
Table 2: unit should be in column 1
Data analysis: OK
Results: Need quantification (for specific comment pl see)
Soil respiration rate: quantification is missing add results and comparative with CK
Figure 3: Resolution should be improved
Soil temperature: results are general statements need quantify data
Figure 3: Check figure no and quality of figure
Soil oxygen content: where is the results data
Figure 5 Resolution should be improved
Soil water content: Quantification is missing
Figure 6 Resolution should be improved
Soil bacterial biomass and Root biomass: Quantification is missing
Figure 7: improve figure quality
Table 3 : ok
Table 4: what is fruit in case of maize it should be cob (pl cross check)
Correlation analysis: add results data and R2
Figure 9: need to define a-f figures in title.
Discussion: whole discussion needs restructure and update with latest literature, its look like general statements.
Conclusion: ok
References: pl make correction according journal format.
Author Response

(The authors gave the same response as above.)

Reviewer 3 Report
This study examines the effects of aerated irrigation on soil respiration and yield of maize 2 root zone. The manuscript was quite well prepared and I would think it is nearly ready for publication. Just have some minor comments:
- Section 1: Please include further detailed literature reviews of previous studies on related topics.
- Line 151: what is SDI?
- Line 87: 2160 h -> hours; 350 d => days
- I would suggest to use another abbreviation for aerated irrigation. AI is well known as artificial intelligence.
- There are lots of spelling, grammar and presentation errors across the manuscript. Please carefully check.
Author Response
Dear reviewers:
Thank you for your letter and for the reviewers'comments concerning our manuscript entitled “The effects of aerated irrigation on soil respiration and yield of maize root zone”(ID:1631071).Those comments are all valuable and very helpful for revising and improving our paper,as well as the important guiding significance to our research.We have studied comments carefully and have made correction which we hope meet with approval.Please see attachment.

This manuscript is a resubmission of an earlier submission. The following is a list of the peer review reports and author responses from that submission.
Round 1
Reviewer 1 Report
Please, you can find my comments on the attached file.

Reviewer 2 Report
The manuscript lacks originality, so it is not interesting in the way it was approached. The experiments carried out are of routine characterization of a soil, modifying certain varials. There is poor analysis of the results.
The introduction has no references. How did the authors put together the theoretical framework, set objectives and design hypotheses and their experiments without being guided by previous work?
The conclusions are poor.